# OpenReview forum: "Test-Time Adaptation for Visual Document Understanding"
_ICLR.cc/2023/Conference — Submitted to ICLR 2023_

### Official Review · Reviewer_tuFF · 2022-10-22

**Confidence:** 3
**Correctness:** 3
**Technical Novelty And Significance:** 3
**Empirical Novelty And Significance:** 3
**Recommendation:** 6

**Clarity, Quality, Novelty And Reproducibility:**

Hi, I'm currently on a medical leave and won't be able to perform ICRL review duties. sorry for the late notice.

**Strength And Weaknesses:**

Hi, I'm currently on a medical leave and won't be able to perform ICRL review duties. sorry for the late notice.

**Summary Of The Paper:**

Hi, I'm currently on a medical leave and won't be able to perform ICRL review duties. sorry for the late notice.

**Summary Of The Review:**

Hi, I'm currently on a medical leave and won't be able to perform ICRL review duties. sorry for the late notice.

---

> ### Author Response · Authors · 2022-11-18
> **Response to Reviewer tuFF**
>
> We would like to thank the reviewer for their given score. However, unfortunately due to unavailability of any comment we are not able to address the possible concerns of the reviewer.

---

### Official Review · Reviewer_F7R6 · 2022-10-24

**Confidence:** 4
**Correctness:** 3
**Technical Novelty And Significance:** 3
**Empirical Novelty And Significance:** 3
**Recommendation:** 6

**Clarity, Quality, Novelty And Reproducibility:**

Please see strengths and weaknesses for more details. Availability of promised corresponding code would ideally make the work reproducible. The paper is written clearly in most parts while the proposed idea is original.

**Strength And Weaknesses:**

Strength:
- The paper is well-motivated and clearly written in most parts. This is an interesting problem with the proposed approach very relevant to the community.
- The authors promise the availability of corresponding code and a detailed description of hyperparameters which would be essential for reproducibility if open-sourced.
- Ablation study on domain adaptation (Table 3) for different components of the system is interesting with pseudo labelling playing an important role.

Weakness:

- Some model-generated examples would provide more insight.
- Human evaluation is not provided.
- Even though the results are promising at the initial level, there is a huge performance difference compared to the models which are trained on target data (Table 1 and Table 2). In Table 2, the ANLS score is approx half compared to the train-on-target models.
- It would help to clarify what kind of OCR techniques were used (industrial like GCS or open-sourced models). How would the results compare when different grades of OCR models are used? An ablation study in this regard could definitely help.
- It seems the relevant baselines haven’t been used where the DANN and BN are works from 2015 and this field has evolved a lot in recent years.


Questions:
- Could the authors explain how the results compare with zero-shot models? Is the source-only model used in zero-shot settings in Tables 1 and 2? If yes, it would help to clarify that in the text.
- In continuation of the point above, when it is mentioned on Page 5, that the closed-set assumption is used, how would the zero-shot model with no assumption like this compare to the proposed approach?
- Could the authors clarify what ECE means on Page 6?


Suggestions/Comments:
- It would help to explain DocTTA right at the beginning in the abstract itself so that the reader knows what to expect in the rest of the paper and not just Section 3.
- It is unclear where Train-on-target used the same model architecture (LayoutLMv2BASE) as the source-only and DocUDA/DocTTA methods. It would help to clarify this in the text.
- An example (maybe pictorial) of the layout XB (6-dimension vector) on Page 5 could definitely help articulate the method.


**Summary Of The Paper:**

This paper tackles the task of domain shift in Visual Document Understanding (VDU) involving entity recognition, key-value extraction, and document visual question answering. DocTTA (document test time adaption) method is introduced which takes a pre-trained model from the source domain and adapts it to the target domain using self-supervised learning (masked visual language modeling and pseudo labelling).


**Summary Of The Review:**

The proposed idea is well motivated and novel and the experiments show promising directions. However, the paper could be tweaked a bit to provide more detailed descriptions. Please see strengths, weaknesses, suggestions and comments.

---

> ### Author Response · Authors · 2022-11-18
> **Response to Reviewer F7R6. [1/2]**
>
> We would like to thank you for your positive feedback points and appreciating our writing and methods. Below we address your comments/questions in the order they were received:
>
> ```
> Some model-generated examples would provide more insight.
> ```
> We have added some qualitative results to the Section A.6 of the Appendix that exemplify correction cases with DocTTA.
>
> ```
> Human evaluation is not provided.
> ```
> For evaluating the predictions made by DocTTA and other baselines we use F1-score on FUNSD-TTA and SROIE-TTA and ANLS score on DocVQA-TTA. We use human-generated labels for computing these metrics and since we are purely performing discriminative modeling and not generative, we think these metrics can be suffice for evaluation.
>
> ```
> Even though the results are promising at the initial level, there is a huge performance difference compared to the models which are trained on target data (Table 1 and Table 2). In Table 2, the ANLS score is approx half compared to the train-on-target models.
> ```
> Great observation! This shows the severity of the distribution shifts, and also the room for improvement if we have methods that can take advantage of the labeled target data, if they become available. We will add a discussion around this.
>
>
> ```
> It would help to clarify what kind of OCR techniques were used (industrial like GCS or open-sourced models). How would the results compare when different grades of OCR models are used? An ablation study in this regard could definitely help.
> ```
> We have explained in the Appendix Sec A.1.3 that for all the benchmarks, we use the officially-provided OCR annotations released with the original dataset in each benchmark. Here are the results from running our experiments with Google’s Tesseract which is an open-source OCR engine compared to what's reported in Table 3 and 8 (same as Table 3 with STD given in parentheses). Results show very small improvements over using the original ones.
>
> | Source                               | Emails&Letters (E) |             |             |
> |--------------------------------------|--------------------|-------------|-------------|
> | Target                               | F                  | T           | L           |
> | DocTTA w/ public Tesseract OCR              | 39.73(0.23)        | 36.02(0.54) | 49.86(2.31) |
> | DocTTA w/ original OCR (Table 3 & 8) | 40.36(0.53)        | 35.28(0.76) | 49.35(1.20) |
>
> ```
> It seems the relevant baselines haven’t been used where the DANN and BN are works from 2015 and this field has evolved a lot in recent years.
> ```
> For TTA baselines, since there is no method proposed for documents, we had to adapt SoTA TTA approaches from the computer vision literature to documents. Therefore, we chose the baselines that do not depend on heavy augmentation techniques (TENT, BN, and SHOT) because documents are not sensitive to them. For clarity we have added experiments with AdaContrast (Chen et al., CVPR 22) on DocVQA-TTA benchmark which uses self-supervised contrastive learning as used in MoCo (He et al, CVPR 20) combined with pseudo labeling. As shown below, the performance gain of using AdaContrast is insignificant compared to methods that do not combine pseudo labeling with any self-supervision (SHOT) and is significantly outperformed by our DocTTA method.
>
> | Source            | Emails&Letters (E) |       |       |
> |-------------------|--------------------|-------|-------|
> | Target            | F                  | T     | L     |
> | AdaContrast (TTA) | 37.21              | 27.43 | 38.69 |
> | DocTTA | 40.36              | 35.28| 49.35 |
>
>
>  In unsupervised domain adaptation settings, we followed the same reasoning for not using augmentation-based methods despite being more recent. However, here we show the more recent baseline SHOT (ICML 20) adapted to the UDA setting which is outperformed by our DocUDA as shown below:
>
> | Source            | Emails&Letters (E) |       |       |
> |-------------------|--------------------|-------|-------|
> | Target            | F                  | T     | L     |
> | SHOT (UDA)        | 39.02              | 31.35 | 48.87 |
> | DocUDA            | 39.23              | 43.54 | 57.99 |
>
> All the above results are added to the Sec A.7 of the Appendix.
>
> ```
> Could the authors explain how the results compare with zero-shot models? Is the source-only model used in zero-shot settings in Tables 1 and 2? If yes, it would help to clarify that in the text.
> ```
> Zero-shot classification typically refers to the ability of a model to recognize new classes whose instances may not have been seen during training. However, in closed-set TTA source and target domains have complete class overlap but they contain samples from different distributions.
>
>
> ```
> Could the authors clarify what ECE means on Page 6?
> ```
> Thank you for pointing that out. It is Expected Calibration Error which is a metric to measure calibration of a model. We added a footnote to describe it. We also have a section devoted to it in the Appendix (Sec A.3).

---

> ### Author Response · Authors · 2022-11-18
> **Response to Reviewer F7R6. [2/2]**
>
> ```
> It would help to explain DocTTA right at the beginning in the abstract itself so that the reader knows what to expect in the rest of the paper and not just Section 3.
> ```
> Thanks for the suggestion, we have “We propose DocTTA, a novel test-time adaptation method for documents, that does source-free domain adaptation using unlabeled target document data” in the second line of the abstract.
>
>
> ```
> It is unclear where Train-on-target used the same model architecture (LayoutLMv2BASE) as the source-only and DocUDA/DocTTA methods. It would help to clarify this in the text.
> ```
> They all use the LayoutLMv2BASE model pre-trained on the IIT-CDIP dataset. This is explained in A.2.1 of the Appendix along with other details of the implementation.
>
>
> ```
> An example (maybe pictorial) of the layout XB (6-dimension vector) on Page 5 could definitely help articulate the method.
> ```
> We have added a figure on this in Sec A.8 of the Appendix  that shows a bounding box associated with each word in the text input sequence and is represented with a 6-dimensional vector in the form of $(x_{min}, x_{max}, y_{min}, y_{max}, w, h)$. where $(x_{min}, y_{min})$ corresponds to the position of the lower left corner and
> $(x_{max}, y_{max})$ represents the position of the upper right corner of the bounding box and $w$ and $h$ denote the width and height of the box, respectively.
>
> ```
> Availability of promised corresponding code would ideally make the work reproducible. The paper is written clearly in most parts while the proposed idea is original.
> ```
> Thanks for the feedback. We will release the code upon acceptance of the paper.

---

> ### Comment · Reviewer_F7R6 · 2022-12-02
> **Rebuttal update**
>
> I have read the author's response and I feel most of my comments have been addressed. I stand by my original ratings.
>
> Additionally, I would suggest the authors to refer the Appendix discussions in the main paper.
>
> Thanks!

---

> > ### Author Response · Authors · 2022-12-02
> > **Thank you for your reply**
> >
> > We would like to thank you for taking the time to read our response. We will refer to the Appendix in the main paper as you suggested in the final version of the paper.

---

### Official Review · Reviewer_eUpw · 2022-10-24

**Confidence:** 3
**Correctness:** 2
**Technical Novelty And Significance:** 2
**Empirical Novelty And Significance:** 2
**Recommendation:** 5

**Clarity, Quality, Novelty And Reproducibility:**

The paper is clear and well written. The novelty of the paper is not clear specially with respect to existing TTA methods. Main novelty could be in integrating the specific MVLM for the case of TTA in documents. Code will be released upon publication for reproducibility.

**Strength And Weaknesses:**

Strengths
- The paper addresses a setting which has not much explored in document understanding, transfering a model learned in a source domain to a target domain. This is a relevant problem in document understanding due to privacy issues or difficulty in getting annotation for new documents.
- The proposed framework takes advantage of the specificity of document understanding integrating a masked language modeling task in addition to the generation and selectoin of pseudo-labels.
- A new set of datasets specific to the task of test-time adaptation are created and released
- Experiments show that the proposed method help to signifficantly improve the results in the target domain in all cases and that all components of the framework contribute to this improvement.

Weaknesses
- Apart from integrating masked language modelling, the difference of the proposed framework with previous TTA methods is not clear. The difference and contribution with respect to other methods to generate and select pseudo labels should be made more explicit, specially in relation to reference (Rizve et al. 2021). It seems that the contribution of the paper lies mainly on integrating documen specific MVLM.
- The comparison with other methods in the state-of-the-art does not help to evaluate the actual contribution of the proposed method. It is not clear whether better results are due only to the MVLM strategy, to a better selection of pseudo labels, or to a combination of both. I understand that the proposed MVLM strategy could be integrated with most of the existing methods for TTA. Then, it would be more insightful either performing the comparison by also integrating MVLM into these existing methods or evaluating the proposed method without it, in order to decouple the effect of both strategies. Actually, results in tables 2 and 3 seem to show that DocTTA without the specific MVLM would perform worse thant SoA TTA methods
- The dataset split between source and target domains seem, in some of the cases a bit arbitrary and not very realistic. The motivation behind such splits is not well motivated. Why is it appropriate to split FUNSD by text density and SROIE by blur and other artifacts. Why creating new categories in DocVQA that do not correspond to document types? In addition, details on how these splits were created are not provided. Were they created manually or automatically? Which criteria were used to filter or annotate documents in each split? Is there already any difference on the performance of the original full model on each of the splits?
- The splits in the target domain do not contain training, validation and test sets.


**Summary Of The Paper:**

The paper proposes a framework for performing test time adaptation of a model trained in a source domain to a target domain, without use or labeled data in the target domain. The main contribution of the paper is applying this framework to a model for document image representation, leveraging standard masked langage modeling (traditionally used in to learn models for document representation) with a pseudo-labelling selection strategy. To evaluate this new setting in document understanding, new datasets are created by splitting standard document datasets into a source and a target domain. Experiments are performed on these proposed datasets, comparing the proposed method with state-of-the-art domain adaptation and test-time adaptation approaches.


**Summary Of The Review:**

The paper proposes an adaptation of TTA methods to the case of document understanding, where this issue has not been much explored. The proposed method seems to rely on standard tecniques for TTA and integrating the specific MVLM objective when retraining the model in the target domain. In this sense, novelty and signifficance of contribution seems limited. The rationale behind the splits for each dataset is not well motivated and comparison with SoA should take into account somehow the effect of specific MVLM task. Overall, I consider that it is not clear the contribution of the paper both methodologically and experimentally.

---

> ### Author Response · Authors · 2022-11-18
> **Response to Reviewer eUpw [1/3]**
>
> We would like to thank you for your positive feedback points and appreciating our methods, datasets and results, particularly acknowledging that our proposed method DocTTA “addresses a not-much-explored setting” that can “significantly improve the results in the target domain in all cases”.
>
> Here we address your comments/concerns in the order they have been raised:
>
> ```
> Apart from integrating masked language modelling, the difference of the proposed framework with previous TTA methods is not clear. The difference and contribution with respect to other methods to generate and select pseudo labels should be made more explicit, specially in relation to reference (Rizve et al. 2021). It seems that the contribution of the paper lies mainly on integrating documen specific MVLM.
> ```
> To better clarify the novelty of the proposed method compared to DocTTA literature, we have added extra explanations into the Introduction section. Overall, DocTTA distinguishes from the other TTA methods as:
>
> - DocTTA is the only TTA approach that combines multimodal self-supervised representation learning with pseudo labeling. Other pseudo labeling based TTA approaches proposed in image classification task, either do pseudo labeling alone (eg. SHOT) or combine contrastive learning as the self-supervision with pseudo labeling (eg. AdaContrast by Chen et al, CVPR 22).
>
> - DocTTA also introduces a new way to select and filter pseudo labels for the purpose of TTA for documents. Unlike SHOT method that does not filter pseudo labels, DocTTA proposes using Shannon’t entropy as a measure of uncertainty to select certain pseudo labels only. This is also different from the paper by Rizve et al. 2021 (see the last six lines of page 5), which focuses on performing semi-supervised learning at computer vision tasks (image and video classification) with pseudo labeling such that they can bridge the performance gap between consistency regularization methods and pseudo labeling based methods. For that, they show a pseudo-labeling selection mechanism that relies on both confidence (softmax output probabilities) and uncertainty (measured by MC-Dropout), performs the best. In our paper, the focus is on test-time adaptation in a multimodal setting (not just image data but language and layout information) for which we use pseudo labeling (along with MVLM and diversity loss) and we show that the most effective pseudo labeling selection mechanism is the one that relies on uncertainty measured with Shannon’s entropy only.
>
> ```
> The comparison with other methods in the state-of-the-art does not help to evaluate the actual contribution of the proposed method. It is not clear whether better results are due only to the MVLM strategy, to a better selection of pseudo labels, or to a combination of both. I understand that the proposed MVLM strategy could be integrated with most of the existing methods for TTA. Then, it would be more insightful either performing the comparison by also integrating MVLM into these existing methods or evaluating the proposed method without it, in order to decouple the effect of both strategies. Actually, results in tables 2 and 3 seem to show that DocTTA without the specific MVLM would perform worse thant SoA TTA methods
> ```
> We have already performed extensive ablation studies in Table 3 to show the impact of different loss components in DocTTA, by including removing MVLM, pseudo labeling, and diversity loss terms. We also show the impact of the pseudo labeling selection mechanism (using confidence or uncertainty or both). Table 3 shows that pseudo labeling selection mechanism has the largest effect on results which is actually a missing point in computer vision SoTA TTA methods such as SHOT where they do not use any selection mechanism and simply train with all the obtained pseudo labels. We also showed that MVLM loss and diversity maximization criteria have similar impact on DocTTA’s performance. Overall, all of the proposed components play important roles in the strong performance, and our method is designed to combine them in a balanced way. Also results for DocTTA w/o MVLM compared to SHOT on E→F and E→T shows to be 3.32% and 1.83% worse, respectively while on E→L is 6.73% better.

---

> ### Author Response · Authors · 2022-11-18
> **Response to Reviewer eUpw [2/3]**
>
> ```
> The dataset split between source and target domains seem, in some of the cases a bit arbitrary and not very realistic. The motivation behind such splits is not well motivated. Why is it appropriate to split FUNSD by text density and SROIE by blur and other artifacts. Why creating new categories in DocVQA that do not correspond to document types?
> ```
> First and foremost, we note that the real-world dataset shifts for document data can be very diverse. Our primary motivation with the proposed benchmarks was to propose a diverse evaluation suite that can reflect real-world performance of the adaptation methods - that’s why we used different datasets and different ways of splitting criteria. The proposed shifts are also inspired by what we have empirically found in our experiences from real-world document data  that causes performance degradation between training and test time. Visual appearance discrepancy (ink color, additional hand written text, etc), different content and/or layout, or sparsely filled templates are the artifacts that we have found documents are not able to generalize to. In particular, the datasets differ in what characteristics they show the largest variation on. For FUNSD vs SROIE datasets, different properties create more substantial differences in split distributions. The reason we used FUNSD for sparsity is that this dataset is known for being noisy due to sparse information and we used SROIE because we noticed the visual discrepancy among its receipts. In DocVQA we searched for keywords matching with the exact desired content in the document (tables, or figure, or email, etc).
>
> Lastly, we also note that we have tried our DocTTA method on other real-world datasets with real-world shifts (that are not included in the paper, as the paper focused on the results on publicly-available datasets) and we observed similar performance lifts.
>
>
> ```
> Is there already any difference on the performance of the original full model on each of the splits?
> ```
> Here is the performance on FUNSD and SROIE original train and test splits used as the source and target domain respectively. DocTTA on the original train and test splits improves source-only results by 1.96% and 0.56% on FUNSD and SROIE, respectively whereas it improves those by 3.43% and 1.89% on our proposed FUNSD-TTA and SROIE-TTA benchmarks, respectively. We believe that this is due to the larger domain gap that exists in our proposed benchmarks compared to the original train and test splits of the datasets.
>
> |                                               | FUNSD   | SROIE   |
> |-----------------------------------------------|---------|---------|
> | Source-only                                   | 82.67  | 96.25  |
> | DocTTA on original splits                     | 84.63  | 96.81  |
> | Performance gain by DocTTA on original split  | +1.96 | +0.56 |
>
> ```
> In addition, details on how these splits were created are not provided. Were they created manually or automatically? Which criteria were used to filter or annotate documents in each split?
> ```
> In Sections 4.1, 4.2, and 4.3 we describe in detail how the splits are created. Statistics of the splits are also given in Appendix section A.1.2.. In summary, for FUNSD-TTA benchmark, we combine the original training and test splits and then manually divide them into two groups such that documents filled with more text were chosen for the source domain. This resulted in having 149 samples for the source and 50 samples for the target domain. For SROIE-TTA benchmark, we also first combined the original training and test splits and then manually divided them into two groups based on their visual appearance – source domain with 600 documents contains standard-looking receipts with proper angle of view and clear black ink color and for the target domain, we chose 347 receipts with slightly blurry look, rotated view, colored ink, and large empty margins. And lastly for DocVQA-TTA , since DocVQA dataset doesn’t have public meta-data to easily sort all documents with their questions, we use a simple keyword search to find our desired categories of questions and their matching documents. We use the same words in domains’ names to search among questions (i.e., we search for the words “email” and “letter” for Emails & Letters domain). However, for Layout domain, our list of keywords is [“top”, “bottom”, “right”, “left”, “header”, “page number”] which identifies questions that are querying information from a specific location in the document.

---

> ### Author Response · Authors · 2022-11-18
> **Response to Reviewer eUpw [3/3]**
>
> ```
> The splits in the target domain do not contain training, validation and test sets.
> ```
> We provide the list of the documents in the source and target domains for our three benchmarks. Files are located at Supplemental/TTA_Benchmarks/. In Section A.1.2 (Dataset Splits) of the Appendix, we have explained that we use a seed number 42 to randomly select 10 and 39 documents for FUNSD-TTA and SROIE-TTA benchmarks, respectively. We used the same seed to select the validation set on all the four domains of DocVQA-TTA as well. The validation set size in DocVQA-TTA domains are given in Table 7 in the Appendix. We would like to highlight again that the validation set samples are selected from the source distribution in all the benchmarks.
>
> ```
> Clarity, Quality, Novelty And Reproducibility:
>
> The paper is clear and well written. The novelty of the paper is not clear specially with respect to existing TTA methods. Main novelty could be in integrating the specific MVLM for the case of TTA in documents. Code will be released upon publication for reproducibility.
> ```
> We have clarified how our proposed TTA method differentiates above, and improved our Introduction section to reflect that. We hope this solves your concerns on the novelty.
>
> ```
> Response to the points raised in the ```Summary of the Reviewer```:
> ```
>
> Please see our explanations above on the rationale behind datasets. We hope these resolve your concerns. We expect that our proposed DocTTA datasets will help accelerate the research on real-world challenges of document understanding and its widespread adoption.
> The contributions of our paper can be summarized as framing the TTA problem for document data for the first time, releasing a set of benchmark datasets, and proposing an effective TTA method that outperforms the alternatives significantly.

---

### Decision · Program_Chairs · 2023-01-20

**Decision:**

Reject

**Justification For Why Not Higher Score:**

The empirical studies raise concerns of reviewers as above

**Justification For Why Not Lower Score:**

The test-time adaptation problem and perhaps the method is interesting

**Metareview: Summary, Strengths And Weaknesses:**

The paper studies test time adaptation of a model trained in a source domain to a target domain without labeled data. A method is proposed with components of masked language modeling and pseudo-labeling. For evaluation, the paper splits existing document datasets into source and target domains. Comparisons are made w.r.t SoTA domain adaptation and test-time adaptation approaches. Reviewers have concerns on the empirical results, in particular the contribution of pseudo-labeling (as one of the major novelties claimed in the paper), the performance gap compared to the SoTA approaches, and the discrepancy between the created datasets vs real settings.

**Summary Of Ac-Reviewer Meeting:**

Reviewers expressed concerns about the contribution of pseudo-labeling (as one of the major novelties claimed in the paper), the performance gap compared to the SoTA approaches, and the discrepancy between the created datasets vs real settings.